# Fertility Preservation in Endometrial Cancer—Treatment and Molecular Aspects

**DOI:** 10.3390/medicina59020221

**Published:** 2023-01-24

**Authors:** Melanie Schubert, Liselotte Mettler, Aarti Deenadayal Tolani, Ibrahim Alkatout

**Affiliations:** 1Department of Obstetrics and Gynecology, University Hospital of Schleswig Holstein, Campus Kiel, 24105 Postcode Kiel, Germany; 2Mamata Fertility Hospital, Infertility Institute and Research Centre, Secunderabad 500026, Telangana, India

**Keywords:** endometrial cancer, fertility preservation, molecular characterization

## Abstract

Endometrial cancer is one of the most common gynecological malignancies worldwide; incidences are rising, with 417,367 new cases registered in 2020. Of these, the proportion of women that are of reproductive age is around 4–14% and the number is increasing. Thus, in addition to oncological therapy and safety, the preservation of fertility plays a central role in therapeutic strategies. Molecular genetic patient data provide a robust supplementary benefit that improves primary risk assessment and can help design personalized treatment options to curtail over- and undertreatment and contribute to fertility preserving strategies. The aim of our review is to provide an overview of the latest significant recommendations in the diagnosis and therapy of endometrial cancer during reproductive age. In this paper the most recent groundbreaking molecular discoveries in endometrial cancer are highlighted and discussed as an opportunity to enhance the prognostic and therapy options in this special patient collective.

## 1. Epidemiology

Endometrial Cancer (EC) is one of the four most common malignancies worldwide, with a higher incidence in industrialized countries such as America and Europe. With 10,930 new cases per year, it is one of the most common malignancies in women in Germany and the most common cancer of the female reproductive organs [1]. The frequency of women ≤40 years of age suffering from EC is 4–14% [2,3]. According to the World Health Organization, in 2020 worldwide there were 35,915 cases of EC in women ≤44 years, 14,203 in women ≤39 years and 2232 in women ≤29 years [4].

The incidence of EC is continuously increasing due to the spread of the western lifestyle, which in turn has led to significant increases in contributory factors such as obesity, diabetes mellitus and delayed childbearing [5]. Significantly, Zhang et al. reported a global increase of 0.58% in the age-standardized incidence rate of EC from 1990 to 2017. Moreover, the incidence rate is estimated to increase by more than 50% by 2040 [6].

## 2. Pathophysiology

Currently, we differentiate two types of EC. Type I is estrogen-associated, mostly receptor-positive, endometrioid adenocarcinoma (80–90%) based on hyperplasia with atypia (atypical hyperplasia, AH/endometrial intraepithelial hyperplasia, EIN). Type II is non-estrogen-associated, mostly receptor-negative/weakly positive, usually on an atrophic endometrium or a serous type endometrial intraepithelial carcinoma. This type II comprises serous, clear, and undifferentiated carcinomas (10–20%) [1,7,8]. Type I often features genetic alterations in PTEN inactivation, microsatellite instability, ß-catenin (CTNNB1) and KRAS mutation. In addition, it associates with the hereditary non- polyposis colon cancer (HNPCC, Lynch syndrome). On the other hand, TP53-mutation, E-cadherin inactivation and PIK3CA alteration are seen in type II EC [1,3,7].

## 3. Risk Factors

An uncontrolled estrogen influence on the endometrium is the underlying cause of type I ECs, which are associated with late menopause, early menarche, nulliparity, failure to ovulate or infertility, hormone replacement therapy (HRT) and polycystic ovary syndrome (PCOS). Other known risk factors are tamoxifen therapy, increasing age (>50 years), diabetes mellitus, hypertension and obesity [1,7]. Indeed, obesity alone causes a four-fold increased risk of EC [6]. Endometrioid EC type I arises based on AH and an already existing concurrent EC in women with AH is found in up to 30–40% [7,9].

Ten percent of ECs are based on genetic disease. One commonly hereditary cancer syndrome is Lynch syndrome, an autosomal-dominant disease caused by mismatch repair deficient (MMRd) genes MLH1, MSH2, MSH6, PMS2 and ECAM with a lifetime risk for EC of up to 16–54%. Another much rarer genetic disease with a lifetime risk for EC of up to 19–28% is Cowden syndrome, an autosomal-dominant disease caused by mutation of PTEN and is also known as PTEN hamartoma tumor syndrome [1].

On a molecular genetic level, based on a genome-wide analysis, the Cancer Genome Atlas (TCGA) differentiates EC into four prognostically significant groups: polymerase ε-mutant/ultramutated (POLEmut), microsatellite instability/hypermutated (MSI/MSI-H or MMRd group), copy number low/microsatellite stable/p53-wild-type (p53wt) and copy number high/p53-mutant (p53abn)/‘serous-like’ [10,11].

POLE-mutated tumors often appear phenotypically as high-grade tumors with morphologic heterogeneity and are represented in type I EC. MSI-H tumors are associated with sporadic aberrations or germline alterations, located in the lower uterine segment, and are also represented in type I EC. Copy number low tumors are characterized with low grade endometrioid adenocarcinomas. These tumors are without a specific driver mutation (no specific molecular profile, NSMP group) and are also seen in type I EC, whereas copy number high tumors are characterized by p53-mutation, comprise all serous cancers, some high-grade endometrioid adenocarcinomas and clear cell carcinomas, and are represented in type II EC [1,12]. A correlation with progression-free survival is known and a prognostic value can be shown in the TCGA subgroups regarding a risk profile. Excellent prognosis is seen in POLE-mutated tumors, and intermediate prognosis in MSI-H and copy number low tumors; meanwhile, copy number high tumors correlate with poor outcomes.

The current joint guidelines of the European Society of Gynecological Oncology (ESGO), the European Society for Radiotherapy and Oncology (ESTRO) and the European Society of Pathology (ESP) have already included the molecular diagnostics and have adjusted the surgical and adjuvant therapy recommendations based on the prognostic risk groups. This novel risk stratification model includes molecular TCGA subgroups in association with classic, familiar, clinicopathologic prognostic factors of EC, such as myometrial invasion, histopathologic type and lymph vascular space invasion (LVSI) [11,13]. This risk stratification model has been summarized by Crosbie et al. recently in 2022, Figure 1 [14].

Based on the TCGA, different classifiers were introduced to reduce the costs and difficulties of sequencing analysis in routine clinical practice. One of these promising novel molecular classifiers, based on a combination of immunohistochemistry (IHC) for MMR proteins, sequencing for POLE exonuclease domain mutations (EDMs) and IHC for p53, is performed by the Proactive Molecular Risk Classifier for EC (ProMisE). An algorithm of the ProMisE is demonstrated in Figure 2 [15]. The diagnostic accuracy of IHC for MMR proteins as surrogate for MSI molecular testing in EC has been calculated by Raffone et al. (2020) in a systematic review [16]. Testing for MMR status/MSI already provides a value by defining the pathological type, by identifying risks for genetic diseases (such as Lynch syndrome), by predicting a prognosis (inspired by the TCGA) and by offering a targeted treatment with immune checkpoint inhibitor therapy [11]. It has also been determined that MMRd is a highly specific predictor of recurrence of AH/EC after initial regression [17]. However, in the setting of endometrial hyperplasia routine immunohistochemical analysis of MMR proteins should not be performed at the current time [18].

Sixty percent of patients with MMRd are known to be carriers of a germline variant of MMR genes. Therefore, screening for a germline variant of MMR genes is essential in this young population and should be performed in all endometrial carcinomas [11,19]. The German S3-guideline on EC published in 2022 has also included the importance of immunohistochemical determination of MMR proteins and recommends that every diagnosed EC should be investigated for MMR defect/MSI regardless of age and histological subtype. This predictive testing should already be performed on the curettage [18].

Some of the already mentioned markers can also be detected by panel-based next-generation sequencing (NGS), a procedure that has not been routinely established in the clinical diagnostics of EC, but is already used in the diagnosis of hereditary cancer syndromes [20,21,22]. Table 1 summarizes a gene panel of the most commonly tested genes identified by next-generation sequencing.

In addition to these named classifiers, there are even more relevant prognostic biomarkers, including CTNNB1 mutation status, estrogen and progesterone expression, amplification of 1q32.1, LVSI, or L1 cell-adhesion molecule (L1CAM) overexpression [23]. L1CAM, for example, proved to be a significant indicator of high-risk disease in EC and is most frequent in p53 abnormal tumors (80%). Studies have shown that L1CAM is predictive of worse outcome among tumors with no specific molecular profile (p53wt/NSMP). Moreover, L1CAM has a significant correlation to distant recurrence and a significant prognostic impact for disease-specific survival [10,24]. Recently, Raffone et al. (2022) proposed LVSI as a prognostic indicator independent of TCGA signature, which increases the risk of mortality and recurrent or progressive disease by 1.5–2 times [25].

In premenopausal women, EC is primarily present as an early stage and with a well-differentiated grade 1, which is associated with the best prognosis. In addition to these clinical-pathological parameters, the mentioned molecular risk profile can be used as an additional prognostic factor for deciding further treatment. Studies have already reported that molecular classification of EC and AH prior to conservative management is reasonable and may predict patients at risk of tumor progression. Puechl et al. (2021) demonstrated the performance of molecular classification using ProMisE for initial diagnosis prior to the administration of a levonorgestrel-releasing intrauterine system (LNG-IUS) for cancer or AH. In addition, they were able to identify each of the molecular subgroups and the resulting risk probability for tumor progress. Patients with p53abn tumors were demonstrated to have the worst outcome with the highest rate of progression or requirement of definitive therapy (50%), whereas POLE-mutated tumors progressed or required definitive therapy in just 25% of cases and showed the longest median time to progression [26].

## 4. Clinic

EC is detected early due to bleeding disorders, such as abnormal premenopausal and postmenopausal bleeding. Physical examination followed by transvaginal ultrasound (TVUS) has 78–85% sensitivity and 82–84% specificity for detecting myometrial invasion [27]. A definitive diagnosis is made by biopsy, usually hysteroscopic guided, followed by fractional curettage (sensitivity of 99.2% and specificity of 86.4%; gold standard), Tao brush cytology, or Pipelle (positive predictive value of 81.7% and negative predictive value of 99.1%, if adequate samples are taken) [1,7].

Studies have shown that blind dilatation with fractional curettage (D&C) has a high risk of resulting in undiagnosed EC (32.7%). Less than half of the uterine cavity is evaluated with blind D&C in 60% of cases [28]. Hysteroscopy with directed biopsy/curettage is more effective in diagnosing cervical involvement (specificity 98.71% vs. 93.76% (*p* < 0.01)) [29,30] and more accurate in the diagnosis of EC histology type and tumor grade than blind D&C [31]. Different algorithms are defined, for example by the German interdisciplinary S3-guidelines on EC published in April 2018 and updated in September 2022 (Figure 3 and Figure 4). In cases of premenopausal abnormal uterine bleeding with hemodynamic relevance, hysteroscopy and fractional curettage represent the therapy of choice. In premenopausal abnormal uterine bleeding with no hemodynamic relevance, clinical investigation and anamnesis with a focus on risk factors, cytology and TVUS are recommended. Pathological causes such as fibroids, disturbed early pregnancies and cervical pathologies are thereby initially excluded. If the examinations reveal an endometrium with thickness >20 mm, inhomogeneity, demarcation, polyps or risk factors such as BMI > 30, diabetes mellitus, Lynch syndrome or a suspicious cytology, hysteroscopy and fractional curettage should be performed. In the absence of sonographic criteria of malignancy or other risk factors conservative hormone therapy should be tried. In cases of therapy failure hysteroscopy and fractional curettage are recommended. Deviating from this, a perimenopausal or postmenopausal bleeding with a thickness of the endometrium >3 mm (or >5 mm after HRT) or focal thickness >3 mm and ≤5 mm is considered a criterion for malignancy and should be clarified with hysteroscopy and fractional curettage. Peri- or postmenopausal bleeding with a plane and homogeneous endometrium ≤3 mm can be clarified histologically by Tao brush or Pipelle or can be clinically controlled after 3 months and clarified histologically by hysteroscopy and fractional curettage in case of persistence [1].

## 5. Staging

After the histological diagnosis has been made, further investigations are performed to rule out metastasis and other concurrent cancers such as ovarian cancer, and to determine the tumor stage. In addition to the medical history, family history and clinical examination, TVUS is performed for exclusion of malignancies in the ovaries, ascites and the mapping of myometrial infiltration. Magnetic resonance imaging (MRI) can be used additionally to determine myometrial and cervical infiltration (sensitivity from 81–90% and specificity from 82–89% [27]). Instead of pelvic MRI, expert vaginal ultrasound examinations can be used to detect myometrial invasion and cervical stromal invasion. As shown in Figure 5, sagittal and coronal transvaginal views of the uterus in 2D and 3D can show details of malignancy such as irregular thickness, a poorly defined endometrial midline, suspicious perfusion and vascular pattern [11]. The International Endometrial Tumor Analysis (IETA) group summarized a description of the sonographic features of the endometrium and intrauterine lesions [32]. Figure 5 shows different imaging of EC stage I in 2D and 3D TVUS.

The low sensitivity (83%) and specificity (42%) of computed tomography (CT) means that it is not well suited for assessing myometrial involvement or cervical invasion [33]. However, CT scans of the chest/abdomen/pelvis are the gold standard to rule out extrauterine spread, lymphadenopathy and metastasis [1,7,11].

A highly sensitive and specific imaging method,18-flurodeoxyglucose positron emission tomography-computed tomography can detect recurrence (89.5–95% and 91–96.4% [33,34,35]) and distant metastases (100% and 96% [33]). Considering that imaging is still a poor way to detect lymph node metastasis, accurate surgical staging is important. Table 2 shows the different diagnostic assessments for EC with their detection value.

Diagnostic laparoscopy can be performed to rule out endometrial cancer outside myometrium or accompanying ovarian malignancies. Because of the relatively low incidence of accompanying ovarian cancer (4.5%), a diagnostic laparoscopy is not mandatory when based on good data in low-risk early EC, no myometrium invasion, grade 1 endometrial EC, unsuspicious ovaries and normal CA-125 [40]. The final classification based on operative staging of EC is made by the Tumor–Node–Metastasis (TNM) Classification and the International Federation of Gynecology and Obstetrics (FIGO) shown in Figure 6 [7].

## 6. Conventional Treatment

The treatment of EC is based on the tumor stage and consists of surgical and non-surgical treatment. The classic surgical therapy of EC includes total hysterectomy, either abdominal or preferred minimally invasive by conventional laparoscopy or robotic assisted surgery with or without bilateral salpingo-oophorectomy (BSO), peritoneal lavage and, if necessary, pelvic and paraaortic lymphadenectomy (LNE).

A systematic pelvic- and paraaortic LNE is recommended in stage pT1b grade 3 (G3) to pT4 in type I EC with the intention of R0-Resection. In type II EC, systematic pelvic and paraaortic LNE is recommended in every stage. Sentinel lymph node (SLN) mapping instead of systematic LNE has been recommended by the National Comprehensive Cancer Network guidelines since 2014 to improve the detection of lymph node metastases, limit the extent of surgery and thereby reduce morbidity such as may be caused by lymphedema. The pathological detection can be optimized by ultrastaging of the lymph nodes, by multiple sections, routine staining and IHC for epithelial markers. In stage pT1a, G3 and pT1b G1/2 type I EC SLN-biopsy with indocyanine green (ICG) can be performed as a part of controlled studies [12,42]. The ESGO/ESTRO/ESP guidelines recommend, based on the definition of prognostic risk groups, SLN-biopsy in patients with low-risk or intermediate-risk disease. However, in patients with intermediate-risk or high-risk disease surgical lymph node staging should be performed [11].

Through surgical staging an accurate diagnosis, extension of the disease, a prognostic assessment and patients who require further adjuvant therapy can be defined. Radiotherapy with brachytherapy, external beam radiation (EBRT) and the combination of both, or chemotherapy with carboplatin AUC 5-6 plus paclitaxel 175 mg/m^2^ are the common adjuvant therapies to lower the risk of tumor recurrence. No adjuvant treatment is recommended for patients with low-risk EC, or stage I-II POLEmut EC. If there is a higher risk of recurrence, adjuvant brachytherapy is recommended to decrease vaginal recurrence. Therefore, adjuvant brachytherapy is considered for high-grade LVSI negative and for stage II grade 1 endometrioid carcinomas. Additional adjuvant chemotherapy is considered in patients with high-grade and/or LVSI positive EC. Adjuvant EBRT should be performed in high- and intermediate-risk disease with LVSI and/or stage II EC. ERBT with adjuvant or concurrent or sequential chemotherapy and radiotherapy is recommended in high-risk EC. However, there are still groups, for example stage IA non-endometroid carcinomas with myometrial invasion, where adjuvant therapy must be considered individually [11].

## 7. Fertility Preservation Treatment

In fertile women with EC who have a concrete desire to bear a child, organ preservation is justifiable in cases of AH or endometrial carcinoma grade 1 without myometrial invasion and without genetic risk factors.

Knowledge of oncological safety is based on the understanding that the survival of patients with EC stage IA after hysterectomy is equivalent to those after fertility preservation by medical therapy [43]. In addition, it is based on the knowledge that in cases of therapy failure subsequent disease progression is rare, often well-differentiated (G1), confined to the endometrium and so still curable with definitive surgical therapy [2]. The diagnosis of EC should be confirmed by an experienced gynecological pathologist combined with an expert ultrasound examination or MRI of the pelvis [11,40]. The therapy can be conducted in these cases with local removal of the tumor by hysteroscopic resection with a cutting loop electrode and subsequent hormone therapy [44].

Progesterone receptors are more likely present in these well-differentiated tumor cells which leads to a response to progestin therapy [2]. However, there are no predictive markers of progestogen resistance available at the current time [45]. The following options are available: LNG-IUS (52 mg) with or without gonadotropin-releasing hormone receptor agonists (GnRH-agonists), oral progesterone such as medroxyprogesterone acetate (400–600 mg/day), or megestrol acetate (160–320 mg/day). There is no consensus on the exact protocol, dose and duration of therapy [7,40]. Hysteroscopic resection followed by progestin therapy, such as LNG-IUS with downregulation of the ovarian activity by using GnRH-agonists, has the lowest recurrence rate [11,40,46]. Metformin added to these therapies seems to improve overall survival and relapse-free survival in EC patients [40]. These treatment options should be based on an informed consensus and with prioritization of patient autonomy, while nevertheless considering oncological certainty [47].

A total hysterectomy, which leads to an almost 100 percent healing rate, must always be offered to the patient. Explicit information about the conscious withdrawal from a curative therapy with potentially lethal consequences from tumor progression or metastases must be provided. If fertility preservation is requested by the patient in favor of a specific desire to bear a child, the patient should be seen by a specialist for reproductive medicine about her chances to conceive and bear a child. It must be clarified that even with an oncological certainty a potential lethal risk is present and ultimately no pregnancy may occur. The patient must consent to a close checkup and a final hysterectomy in case of treatment failure, recurrence or the completion of family planning. After weighing up the risk factors and the benefits, a therapy regime must be determined together with the patient [11,47].

A regular evaluation for disease regression/recurrence is recommended by imaging and endometrial biopsy via hysteroscope at 3–4 and 6 months. If a recurrence or no response is seen after these 6 months (two negative specimens), the patient needs to proceed with the standard surgical procedure of total hysterectomy [11,40,42]. The reported complete response rate varies closely depending on the stage and grade of the EC. Different therapy protocols depend strongly on the length of treatment and follow-up period. As all studies and meta-analysis are conducted with different therapy protocols and inclusion criteria, it is difficult to come up with comparable statistics for recurrence, regression or pregnancy rates.

A meta-analysis by Gallos et al. reported a pooled regression of 76.2% [48], which is congruent with a meta-analysis by Zhang et al. of 79.5% [49]. Medroxyprogesterone acetate and LNG-IUS combined showed an overall CR of 87.5% [50]. A pooled regression of 94.24% after hysteroscopic resection with LNG-IUS vs. 79.5% with oral progesterone alone is stated by Zhang et al. [49].

The complete response rate increases by the length of treatment, which is shown by Won et al. [40]. The relapse-free survival rate at 5 years is about 73% to 84.8% [51,52]. Pregnancy rates are reported at 61% to 73% [5,51] with a live birth rate of 45% to 66% [46,51] and are reported to be more successful with the support of assisted reproductive technology. De Rocco et al. have recently shown in their systematic review and meta-analysis the highest pregnancy rates (63.1%), lowest miscarriage rates (17.4%) and highest livebirth rates (80.8%) with LNG-IUD compared to megestrol acetate or medroxyprogesterone acetate or GnRH-agonists [53].

There is no uniform recommendation on the duration of the therapy, but it is known that maintenance treatment lowers the recurrence [40]. Therefore, continuous hormonal treatment is recommended until the realization of childbearing. The described therapy algorithm is shown in Figure 7.

The selection of patients suitable for fertility should be chosen carefully by a comprehensive pretreatment evaluation. Basic requirements for fertility preservation treatment are the absence of a contraindication to medical therapy or pregnancy per se [45]. Patients with a poorer outcome, itself due to secondary diseases such as other cancers like breast cancer, stroke, deep vein thrombosis, pulmonary embolism or myocardial infarction, a resulting relative or absolute contraindication to hormone therapy as well as a resulting lower successful birth probability, should be advised very critically with regard to fertility preservation [18,45].

Figure 7 shows the currently recommended basic histopathological optimal indications for this group of patients: grade 1, no myometrial invasion, no genetic factors, p53wt, L1CAM negative [18].

These listed required indications are based on the following knowledge: EC grade 1 and absent myometrial invasion mostly goes along with a very low risk for extrauterine spread [2]. The low probability in this low risk group, regarding the TCGA subgroups risk profile (Figure 1), in combination with the detailed imaging completed before fertility preservation treatment creates a high oncological certainty. Some studies also suggest the possible inclusion of patients with early-stage EC stage IA grade 2 to be considered for fertility preservation. However, this suggestion is not part of the current guidelines based on the limited data and the recognition that pregnancy failure might be related to a higher grade because of the higher PAI-1 level. Larger studies and randomized clinical trials are needed here to support oncological safety in order to recommend this group for fertility preservation [45].

Women with genetic factors such as Lynch syndrome should not go for fertility preservation treatment because of the increased risk to develop other cancers, including ovarian cancer (up to 50%), colorectal cancer (up to 57%) and other cancers like kidney, small bowel and biliary tract cancers. The risk for synchronous and metachronous ovarian cancer is proven to be high in this patient collective [2]. Also, carriers of BRCA1/2 are associated with an increased risk of other cancers, especially for ovarian cancer (up to 44% for BRCA1 and 17% for BRCA2 carriers) and breast cancer (up to 72% for BRCA1 and 69% for BRCA2 carriers) [54].

As previously mentioned, L1CAM, which is most frequent in p53abn tumors, is predictive of a worse outcome, has a significant correlation to distant recurrence and a significant prognostic impact for disease-specific survival. P53abn in stage I EC with no myometrial invasion is classified as an intermediate risk group, regarding the TCGA subgroups risk profile (Figure 1). Therefore, women with a positive detection of L1CAM and p53abn should not receive fertility preservation [10,11,18].

However, other risk factors not listed in Figure 7, but named in the upper chapter for a higher likelihood of treatment failure or even a worse outcome in EC patients, should also be rigorously evaluated and included in the decision-making process as to whether fertility preservation treatment is justifiable. Obesity is known to be a contributing factor for conservative drug failure and a factor affecting the duration of complete response, and must be considered in the decision-making process [2,55].

Preexisting metabolic disorders such as PCOS, insulin resistance, obesity and age can each be the reason for subfertility despite successful hormonal treatment of EC. Therefore, to discuss the chances of pregnancy and to create a treatment plan, it is important to consult a specialist for reproductive medicine before starting the treatment for EC. Optimizing health status with lifestyle interventions is a basic treatment to improve fertility and to reduce the likelihood of tumor recurrence. Being overweight and smoking are the most important factors in need of intervention. Assisted reproductive technology with in vitro fertilization, intracytoplasmic sperm injection, gamete intrafallopian transfer or zygote intrafallopian transfer are demonstrated to be safe with no difference in relapse rates and have a significant improvement on pregnancy rate [55,56,57]. To minimize the risk of tumor recurrence letrozole alone or combined with gonadotropin can be used [57]. However, because of a higher risk of adverse obstetric outcome or obstetric complications, patients have to be monitored closely and have to be aware of their risks [56].

Once family planning has been completed, however, the final therapy should be a hysterectomy [11]. Hysterectomy with ovarian preservation with bilateral salpingectomy can be discussed in premenopausal patients with EC grade 1, without genetic risk factors (e.g., Lynch-Syndrome, BRCA-Mutation etc.) and with myometrial invasion <50% after the previous exclusion of synchronous concomitant ovarian malignancy, ovarian metastasis or extra-uterine disease. There is no significant adverse impact on survival by ovarian preservation in correctly selected cases [3,10,40].

The oncological follow-up of premenopausal patients with EC after a hysterectomy is identical to postmenopausal patients with EC. Patients should be checked up by symptom-oriented anamnesis and a clinical gynecological exam with speculum and rectovaginal palpation every 3 to 6 months during the first 3 years followed by every 6 months during the next 2 years. If there are symptoms, further imaging should be conducted [42]. A secondary BSO should be considered based on a risk-benefit analysis as the data show an increase in coronary heart disease, cognitive impairment, premature death and stroke in premenopausal patients after a BSO without a reduction in the risk of recurrence in the case of an adenectomy in premenopausal patients [40].

## 8. Conclusions

The incidence of premenopausal patients younger than 40 years of age confronted with EC stage 1 and wishing to preserve their fertility continues to rise. An important goal in addition to tumor therapy and oncological safety is to preserve fertility in this selected group. If a patient strongly desires fertility-sparing treatment of EC after full counselling of all potential risks, it can be considered as a safe procedure. Intensive diagnostics and exclusion of risk factors, such as preexisting myometrium invasion, distant manifestation of EC, synchronous malignancies and genetic factors, should be ruled out to choose the correct patients with the lowest disease progression or relapse in the first place. During this process, close multidisciplinary cooperation between the various specialists such as gynecologists, pathologists, geneticists, radiologists and reproductive medicine specialists is essential [54].

A strict continuous follow-up during the treatment to determine response and failure is necessary. Once family planning is completed, the removal of the adnexa should be undertaken.

EC continues to be one of the genital carcinomas with a good prognosis, and novel molecular markers allow us upfront precise diagnosis with definitive prognostic forecast. Therefore, it should be the aim to advance research, collect the rare cases and make progress.

## Figures and Tables

**Figure 1 medicina-59-00221-f001:**
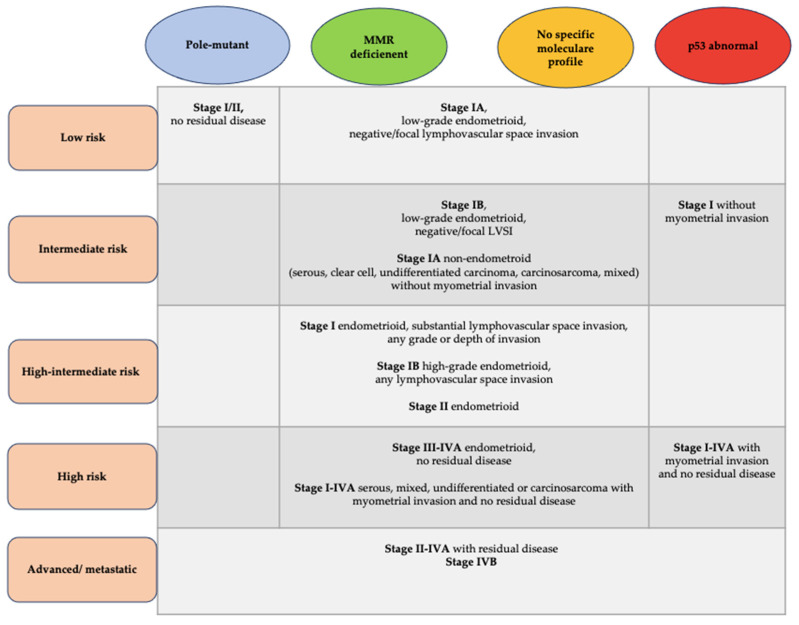
Molecular subgroup classification for defining the ESGO/ESTRO/ESP prognostic risk groups, adapted from Crosbie, E.J. et al. [14].

**Figure 2 medicina-59-00221-f002:**
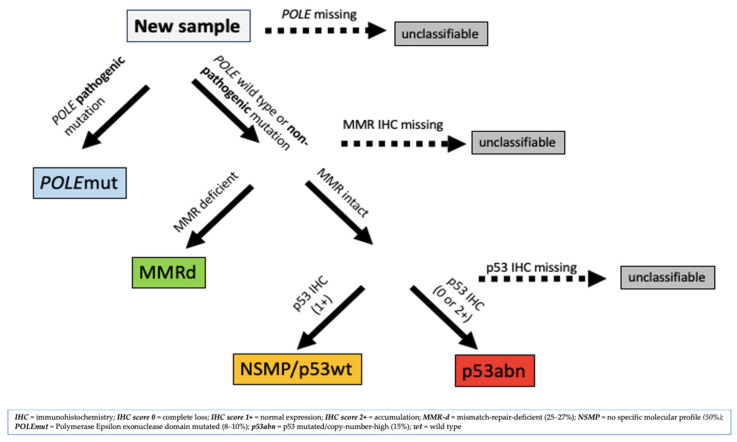
Proactive Molecular Risk Classifier for EC (ProMisE) Algorithm, adapted from McAlpine, J. et al. [15].

**Figure 3 medicina-59-00221-f003:**
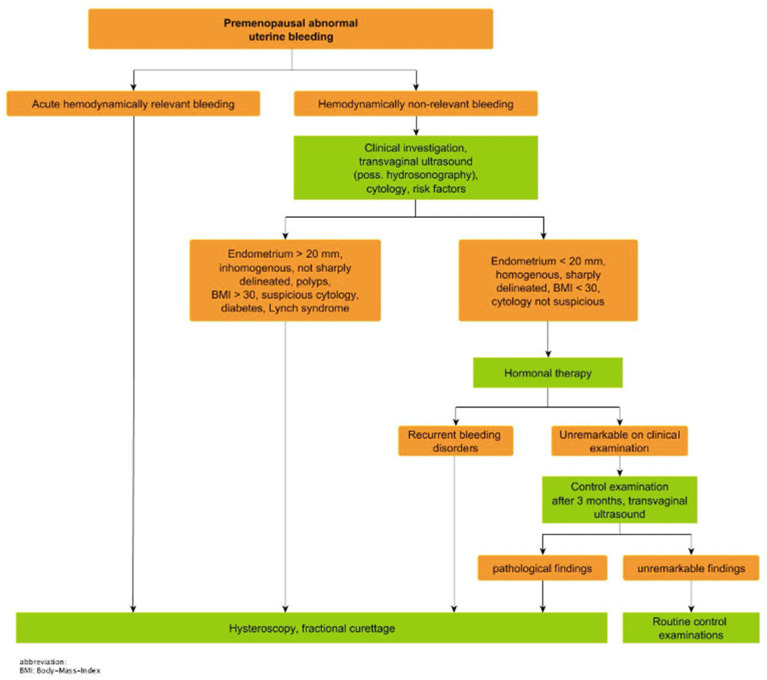
Diagnostic algorithm for abnormal premenopausal uterine bleeding, by Emons G. et al. [1].

**Figure 4 medicina-59-00221-f004:**
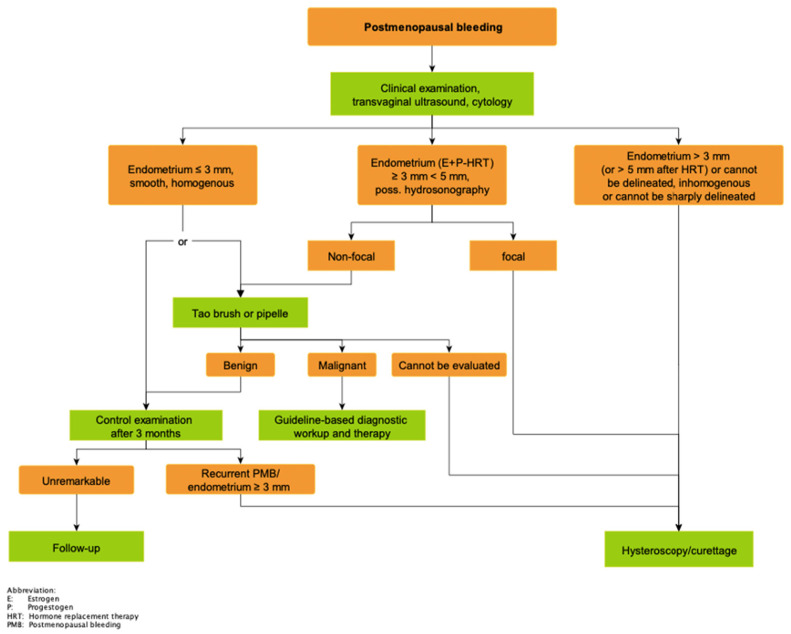
Diagnostic algorithm for the procedure in women with perimenopausal or postmenopausal bleeding, by Emons G. et al. [1].

**Figure 5 medicina-59-00221-f005:**
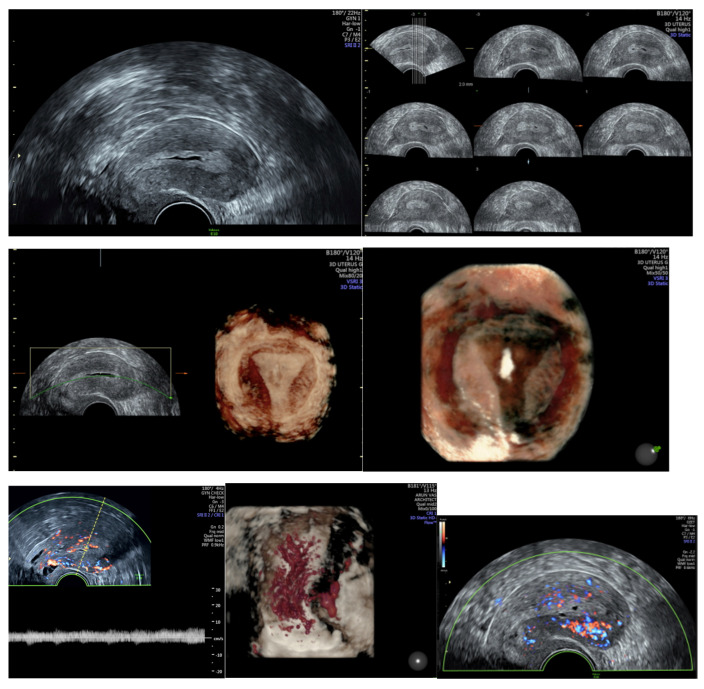
TVUS in endometrial cancer described by the IETA group criteria [32], all pictures by Mamata Fertility Hospital, Secunderabad, Telangana, India. (**a**) TVUS in EC Stage I: sagittal and coronal transvaginal view of the uterus demon-strating a homogeneous myometrium, regular endometrial-myometrial junction, clear differentiation to a hyperechogenic endometrium with irregular thickness, endometri-al folds and intracavitary fluid. (**b**) 3D TVUS in EC Stage I: 3D ultrasound is demonstrating a regular endometrial-myometrial junction, an irregular structure in the right side of the cavity. (**c**) Duplex TVUS in EC Stage I: 3D power doppler in EC Stage I: color score of 4, multiple vessels with multifocal origin, increased vascularization on the right.

**Figure 6 medicina-59-00221-f006:**
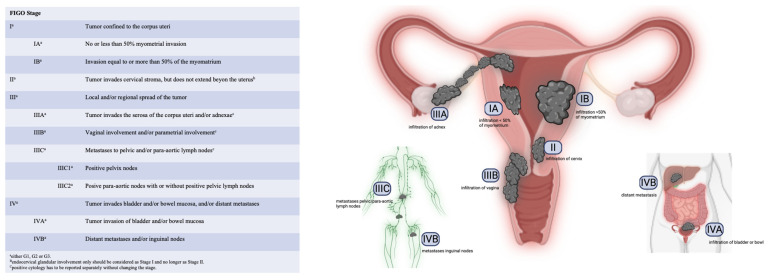
FIGO staging system, created with BioRender.com and adapted from Crosbie E. et al. [14,41].

**Figure 7 medicina-59-00221-f007:**
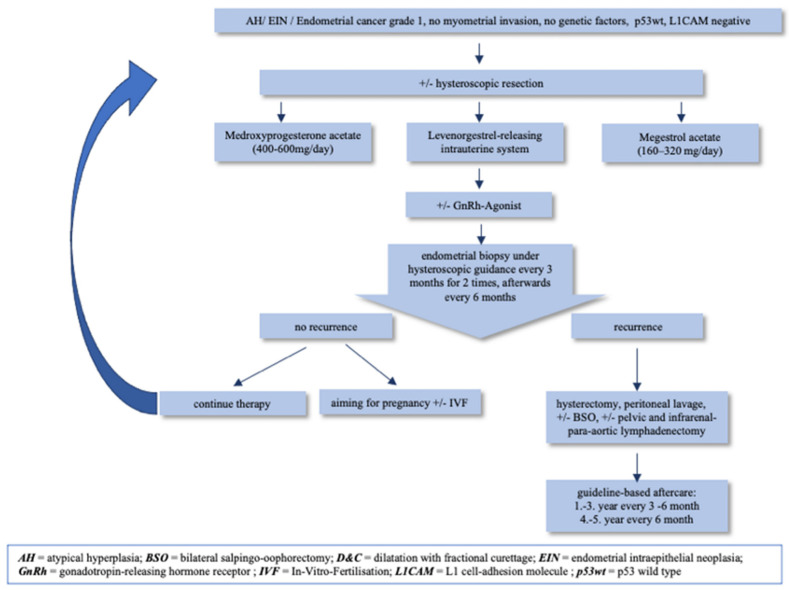
Therapy algorithm of fertility-sparing in selected patients with EC.

**Table 1 medicina-59-00221-t001:** Next-generation sequencing gene panel.

Gene		Chromosome	Methodology	Time	Material	Associated Diseases
EPCAM	epithelial cell adhesion molecule	chromosome 2p21	Next- Generation Sequencing (NGS)	3–6 weeks	2–4 mL EDTA	
MLH1	mutS Homolog 1	chromosome 3p21.3				Lynch-Syndrome, CMMRD-Syndrome
MSH2	mutS Homolog 2	chromosome 2p21p22				
MSH6	mutS Homolog 6	chromosome 2p16				
MUTYH	mutY DNA glycosylase	chromosome 1p34.3–p32.1				
NTHL1	Nth like DNA Glycosylase 1	chromosome 16p13.3				
PMS2	mismatch repair protein 2	chromosome 7p22				Lynch-Syndrome, CMMRD-Syndrome
POLD1	polymerase delta 1	chromosome 19q13.33				
POLE	polymerase epsilon	chromosome 12q24.33				
PTEN	phosphatase and tensin homolog	chromosome 10q23.31				Cowden-Syndrome = PTEN-Harmatoma-Tumor-Syndrome

CMMRD = constitutional mismatch repair deficiency; EDTA = ethylene diamine tetraacetic acid; NGS = Next-Generation Sequencing; POLE = polymerase ε.

**Table 2 medicina-59-00221-t002:** Diagnostic assessment in EC.

	Sensitivity	Specificity	+	-
TVUSmyometrial invasion EC by a cut-off value ≥ 5mm for endometrium thickness	75% [27]–81.6% [36] 96% [33,36]	82% [27]–89.5% [36] 61% [33]	determining the endometrial thickness low cost, great patient tolerability, no contrast agents, available everywhere	myometrial invasion cervical invasion
MRI myometrial invasion cervical invasion lymph node metastases	79% [37]–83% [27] 53% [37]–58% [38] 44% [23]–59% [37]	81% [37]–82% [27] 95% [37,38] 95% [37]–98% [33]	myometrial invasion cervical invasion	
CT myometrial and cervical invasion	83% [33]	42% [33]	extrauterine spread lymphandenopathy metastatic disease beyond the pelvis	myometrial invasion cervical invasion
PET/CT distant metastases lymph node metastases recurrent disease	100% [33] 72% [33,35]–80% [39] 89.5% [34]–95% [33,35]	96% [33] 94% [33,35]–96% [39] 91% [33,35]–96.4% [34]	distant metastasis recurrence	detection of lymph nodes <5 mm

## Data Availability

The datasets analyzed for the current study are available from the corresponding author on reasonable request.

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
