# Peer review of "Fertility Preservation in Endometrial Cancer—Treatment and Molecular Aspects"

_medicina, 2023, doi:10.3390/medicina59020221_

Round 1
Reviewer 1 Report
The presented work is a thorough review of the literature, which contains well-presented factors of the endometrial cancer pathogenesis and a description of the endometrial cancer pathophysiological and molecular biological features. The work presents the classification of molecular subgroups of endometrial cancer in detail , which are associated with a certain prognostic risk of the tumor process progression. The work is well illustrated with diagrams of diagnostic and therapeutic algorithms. The important part of this work is preserve the fertility of women in the reproductive period. Therefore the proposed treatment regimens and monitoring of these patients have the prospect of their further use.
Note: the article contains many abbreviations that are not deciphered, so it is sometimes difficult to understand the presented material.
Author Response
Response to reviewer 1 comments:
Comments and Suggestions for Authors:
The presented work is a thorough review of the literature, which contains well-presented factors of the endometrial cancer pathogenesis and a description of the endometrial cancer pathophysiological and molecular biological features. The work presents the classification of molecular subgroups of endometrial cancer in detail , which are associated with a certain prognostic risk of the tumor process progression. The work is well illustrated with diagrams of diagnostic and therapeutic algorithms. The important part of this work is preserve the fertility of women in the reproductive period. Therefore the proposed treatment regimens and monitoring of these patients have the prospect of their further use.
Note: the article contains many abbreviations that are not deciphered, so it is sometimes difficult to understand the presented material.
Response to Reviewer 1 Comments:
Thank you very much for your interest in this work, for taking your valuable time and for this constructive suggestion for improvement. In our revision of this article, we made a list of abbreviations (Page 14) and avoided unnecessary abbreviations. Please see the attachment.

Reviewer 2 Report
With the spread of the western lifestyle and continuous improvement of endometrial cancer diagnosis, women of childbearing age account for an increasing proportion of endometrial cancer patients, and the fertility preservation in endometrial cancer has attracted more and more attentions widely in the society. In this manuscript, the authors reviewed the epidemiology, pathophysiology, risk factors, clinic, staging, conventional treatment and fertility preservation treatment of endometrial cancer. This is a meaningful paper. The comments are as follows:
1. Please have someone competent in the English language and the subject matter of your paper go over the paper and correct it.
2. I suggest that the authors should add some discussion about the selection of the population for endometrial cancer fertility preservation therapy to fit the general theme better.
3. The authors spent a lot of time introduced the risk factors of endometrial cancer. Whether there are risk factors affect the choice of fertility preservation treatment?
In conclusion, this paper, to be published, needs appropriate revisions in some areas.
Author Response
Response to Reviewer 2 Comments:
Thank you very much for your interest in this work and for taking your valuable time. We accepted and implemented your very productive suggestions for improvement.
Point 1: Please have someone competent in the English language and the subject matter of your paper go over the paper and correct it.
Response 1: The article was again professionally revised in terms of correct English language.
Point 2: I suggest that the authors should add some discussion about the selection of the population for endometrial cancer fertility preservation therapy to fit the general theme better.
Response 2: We have largely revised section “7. Fertility Preservation Treatment” and provided a detailed discussion of population selection for fertility preservation in endometrial cancer (Page 10-12).
Point 3: The authors spent a lot of time introduced the risk factors of endometrial cancer. Whether there are risk factors affect the choice of fertility preservation treatment? In conclusion, this paper, to be published, needs appropriate revisions in some areas.
Response 3: We have accepted your point of criticism and also addressed the relationship of the risks of endometrial cancer and the associated influence on the choice of fertility preservation.

Reviewer 3 Report
Thank you for the opportunity to review this interesting paper.
I think the paper is of a high standard. The structure is good as well as the reference list is complete and updated.
In your opinion, why is there an increase in cases of endometrial cancer in young patients?
I suggest only to take into account this recent paper on fertility strategies in young patients with G2 EEC (doi: 10.1155/2022/4070368).
Author Response
Comments and Suggestions for Authors:
Thank you for the opportunity to review this interesting paper.
I think the paper is of a high standard. The structure is good as well as the reference list is complete and updated.
In your opinion, why is there an increase in cases of endometrial cancer in young patients?
I suggest only to take into account this recent paper on fertility strategies in young patients with G2 EEC (doi: 10.1155/2022/4070368).
Response to Reviewer 3 Comments:
Thank you very much for your interest in this work and for taking your valuable time.
The reason for the increase in cases of endometrial cancer in young patients is multicausal in my opinion: The incidence of endometrial cancer especially in younger patients is continuously increasing due to the spread of the western lifestyle. There has been a significant increase in contributory factors such as obesity, diabetes mellitus and delayed childbearing. In the past women had their children in their 20s, and now they're often 40 years old when it comes to family planning, bringing with them many additional risk factors.
On the other hand, I think especially since gynecologists know about the increased risk of endometrial cancer of their often older, overweight patients (because this is the reality) they are more vigilant. Therefore, the indication for diagnostic workup in bleeding abnormalities is much more generous. Diagnostics have also improved enormously thanks to improved ultrasound equipment and better training. Also, patients with risks and bleeding anomalies are more generously referred to special clinics. Every patient also receives an ultrasound scan as part of the assessment at the IVF clinic. Not only the ultrasound equipment has improved, but also the malignancy criteria and the training of the examiners.
Thanks for your suggestion adding this brilliant recent paper on fertility strategies in young patients with G2 endometrial cancer.
